# Investigation of Anxiety- and Depressive-like Symptoms in 4- and 8-Month-Old Male Triple Transgenic Mouse Models of Alzheimer’s Disease

**DOI:** 10.3390/ijms231810816

**Published:** 2022-09-16

**Authors:** Dorottya Várkonyi, Bibiána Török, Eszter Sipos, Csilla Lea Fazekas, Krisztina Bánrévi, Pedro Correia, Tiago Chaves, Szidónia Farkas, Adrienn Szabó, Sergio Martínez-Bellver, Balázs Hangya, Dóra Zelena

**Affiliations:** 1Center for Neuroscience, Szentágothai Research Center, Institute of Physiology, Medical School, University of Pécs, 7624 Pécs, Hungary; 2Laboratory of Behavioral and Stress Studies, Institute of Experimental Medicine, 1083 Budapest, Hungary; 3János Szentágothai Doctoral School of Neurosciences, Semmelweis University, 1085 Budapest, Hungary; 4Lendület Laboratory of Systems Neuroscience, Institute of Experimental Medicine, 1083 Budapest, Hungary; 5Department of Anatomy and Human Embryology, Faculty of Medicine and Odontology, University of Valencia, 46010 Valencia, Spain

**Keywords:** 3xTg-AD, Alzheimer’s disorder, depression, anxiety, mice models

## Abstract

Alzheimer’s disease (AD) is a progressive neurodegenerative disorder and the most common form of dementia. Approximately 50% of AD patients show anxiety and depressive symptoms, which may contribute to cognitive decline. We aimed to investigate whether the triple-transgenic mouse (3xTg-AD) is a good preclinical model of this co-morbidity. The characteristic histological hallmarks are known to appear around 6-month; thus, 4- and 8-month-old male mice were compared with age-matched controls. A behavioral test battery was used to examine anxiety- (open field (OF), elevated plus maze, light-dark box, novelty suppressed feeding, and social interaction (SI) tests), and depression-like symptoms (forced swim test, tail suspension test, sucrose preference test, splash test, and learned helplessness) as well as the cognitive decline (Morris water maze (MWM) and social discrimination (SD) tests). Acetylcholinesterase histochemistry visualized cholinergic fibers in the cortex. Dexamethasone-test evaluated the glucocorticoid non-suppression. In the MWM, the 3xTg-AD mice found the platform later than controls in the 8-month-old cohort. The SD abilities of the 3xTg-AD mice were missing at both ages. In OF, both age groups of 3xTg-AD mice moved significantly less than the controls. During SI, 8-month-old 3xTg-AD animals spent less time with friendly social behavior than the controls. In the splash test, 3xTg-AD mice groomed themselves significantly less than controls of both ages. Cortical fiber density was lower in 8-month-old 3xTg-AD mice compared to the control. Dexamethasone non-suppression was detectable in the 4-month-old group. All in all, some anxiety- and depressive-like symptoms were present in 3xTg-AD mice. Although this strain was not generally more anxious or depressed, some aspects of comorbidity might be studied in selected tests, which may help to develop new possible treatments.

## 1. Introduction

According to the World Health Organization, Alzheimer’s disease (AD) continues to be the most prevalent form of dementia, contributing 60–80% of cases. In our aging society, this remains a great issue yet to be solved since the most affected population is elderly, whose quantity is increasing.

Up to 50% of AD patients develop anxiety and depression at some stage of the disease, which may promote faster cognitive decline and generally causes greater impairment in the quality of life. A meta-analysis investigating data from 1964 to 2014 found that one of the most common comorbidities with AD is depression: 42% of patients suffered from both [1,2]. Additionally, anxiety symptoms were found to be comorbid in 39% of patients with AD [3]. These neuropsychiatric disorders also produce a very complex group of symptoms, making the diagnoses difficult. Even though the issue has repeatedly been addressed, it remains debated whether depression is a risk factor or a symptom of AD. Although the hypothesis that the treatment of AD by acetylcholinesterase (AChE) inhibitors will increase the risk for depression was not confirmed by [4], we cannot entirely dismiss the possibility of treatment-induced changes, either. Hence, the deeper mechanisms of the comorbidity are yet to be uncovered.

In the ongoing research, preclinical studies play a crucial role in which genetic animal models—especially mouse models—are greatly helpful. The triple transgenic mouse model of AD (3xTg-AD) [5] was considered one of the most relevant in vivo models of AD harboring three human mutant genes: amyloid precursor protein, presenilin-1, and tau protein. In this model, the mutant genes are overexpressing the proteins. Therefore, the appearance of the major pathological hallmarks of AD—β-amyloid plaques and neurofibrillary tangles—are predestined [5,6]. According to previous data, these hallmarks appeared at around 6-month in the 3xTg-AD mice [6]. The progressive cognitive decline was already confirmed in them using tests such as Morris Water Maze (MWM), Barnes maze, and Y-maze [7]. Although selected anxiety- and depression-like behaviors were also examined (Appendix A
Table A1), there was no systematic evaluation, as well as the age of the animals was not sufficiently taken into consideration [8].

The main aim of our study was to investigate whether the 3xTg-AD model was suitable for examining the comorbidity of AD and anxiety and depression. Therefore, the appearance of anxiety- and depression-like comorbid symptoms was studied before and after the appearance of the main pathological hallmarks of AD by comparing 4- and 8-month-old mice with age-matched controls. We assumed that these symptoms would progressively appear (similarly to cognitive decline). However, we could not rule out the possibility that they would precede cognitive decline contributing to the worsening of the symptoms. We confirmed the progressive cognitive decline with a MWM, the “gold standard” in AD research [9,10], and the appearance of structural changes by AChE histochemistry in the somatosensory and motor cortex. A behavioral test battery was used to examine anxiety- (open field (OF), elevated plus maze (EPM), light-dark box (LD), novelty suppressed feeding (NSF), and social interaction (SI) tests), and depression-like (forced swim test (FST), tail suspension test (TST), sucrose preference test (SPT), splash test, and learned helplessness (LH)) changes. The hyperactive hypothalamic-pituitary-adrenocortical (HPA) axis was repeatedly described in depressed patients [11,12], and non-suppression occurred, i.e., they were less sensitive to feedback inhibition of glucocorticoids, the end-hormone [13]. This was the firstly introduced objective psychiatric test, although it worked in only about 64% of patients with major depression [14]. Nevertheless, we mimicked the feedback by a synthetic glucocorticoid, dexamethasone, and examined its efficacy in suppressing the HPA axis.

## 2. Results

### 2.1. Cognitive Tests

#### 2.1.1. Morris Water Maze (MWM) Test

At four months of age, 5 days were sufficient to learn the task as the controls found the hidden platform on the last day of the training during less than 20 s (17.89 ± 1.70 s) (Figure 1A). The 3xTg-AD mice performed a bit slower than the controls, with only a tendency for genotype difference at day 5 (*p* = 0.06). The genotypic difference increased at 8 months of age (F(1,11) = 14.84; *p* = 0.00). In the “probe test”, there was no difference in the distance traveled either between genotypes or age groups, suggesting that locomotion was not impaired (Figure 1B).

#### 2.1.2. Social Discrimination (SD) Test 

The 3xTg-AD animals had poor short-term memory as the discrimination index (DI) was not significantly different from zero (4-month-old: t(7) = 0.01, *p* = 0.99; 8-month-old: t(7) = −0.03, *p* = 0.98; Figure 1C). In contrast, the control group had intact social memory throughout (4-month-old: t(7) = 2.66, *p* = 0.03; 8-month-old: t(5) = 3.54, *p* = 0.02).

### 2.2. Tests of Anxiety-Like Behavior

#### 2.2.1. Open Field (OF) Test

There was a significant difference in time spent in active movement: the 3xTg-AD mice spent significantly less time in active movement compared to the control both in the 4- (F(1,14) = 7.73; *p* = 0.02) and 8-month-old (F(1,12) = 4.87; *p* = 0.04) groups (Figure 2A). Moreover, the 4-month-old 3xTg-AD animals spent more time with self-grooming than the controls, however, the difference was only marginally significant (F(1,14) = 3,28; *p* = 0.09, Figure 2B). At 8-month this difference was further reduced (F(1,12) = 1.51; *p* = 0.24). There were no significant differences in exploration and rearing between either age groups or genotypes.

#### 2.2.2. Elevated Plus Maze (EPM) Test

There was no significant difference between the groups in the time spent in the open arm (age: F(1,33) = 0.48; *p* = 0.49, genotype: F(1,33) = 0.19; *p* = 0.66, age*genotype: F(1,33) = 0.48; *p* = 0.49) compared with the controls (data not shown). There was no significant difference in the open arm ratio, a locomotion-independent measure of anxiety, either (age: F(1,33) = 2.59; *p* = 0.12, genotype: F(1,33) = 0.57; *p* = 0.46, age*genotype: F(1,33) = 0.30; *p* = 0.59; Figure 2C), or in the frequency of entering the closed arm, as a measure of locomotion in this test (age: F(1,33) = 0.01; *p* = 0.94, genotype: F(1,33) = 0.98; *p* = 0.33, age*genotype: F(1,33) = 2.65; *p* = 0.11, data not shown).

#### 2.2.3. Light-Dark (LD) Test

Significant difference was found between the genotypes in percentage of time spent in the dark compartment of the box (genotype: F(1,36) = 10.18; *p* = 0.00; Figure 2D). Namely, the 3xTg-AD mice spent less time in the dark compartment. Moreover, the frequency of stepping into the dark compartment was also significantly lower in 3xTg-AD mice compared with the controls (genotype: F(1,36) = 16.13; *p* = 0.00; Figure 2E). This suggested that 3xTg-AD animals moved less in this test, and their motivation to explore might be lower. However, when we explored the latency of stepping into the dark compartment, not only the age (F(1,36) = 4.21, *p* = 0.04) but also the genotype (F(1,36) = 3.78, *p* = 0.05) and the age*genotype interaction (F(1,36) = 3.86, *p* = 0.05) was significant with genotype difference in the 8- but not in 4-month-old 3xTg-AD animals during the post-hoc test (Figure 2F). Thus, the motivation to explore seemed to gradually decrease.

#### 2.2.4. Novelty Suppressed Feeding (NSF) Test

There was a significant difference in time spent with immobility between genotypes in both age groups (genotype: F(1,43) = 4.54; *p* = 0.03; Figure 2G). More precisely, the 3xTg-AD mice spent more time with this behavior, which suggested enhanced anxiety-like behavior. However, the 3xTg-AD mice spent more time eating as well, without age difference (genotype: F(1,43) = 19.23; *p* = 0.00; Figure 2H). In the latency to first sniff or eat pellet, there was no significant difference between groups (*p* > 0.05, data not shown).

#### 2.2.5. Social Interaction (SI) Test

There was a significant difference between genotypes in the time spent with social behavior, however, this effect was age dependent (age*genotype: F(1,36) = 7.52; *p* = 0.05; 8-month-old post-hoc: *p* = 0.01; Figure 2I). Namely, the 8- but not 4-month-old 3xTg-AD mice spent less time with friendly social behavior compared with the controls. Aggressive and defensive behaviors were not apparent during the test, which could be the reason for not finding significant differences in these parameters.

### 2.3. Tests of Depressive-like Behavior

#### 2.3.1. Forced Swim Test (FST)

A significant difference between genotypes was found in the time spent floating (genotype: F(1,41) = 7.45; *p* = 0.01; data not shown). Interestingly, 3xTg-AD mice spent less time with this behavior compared to the control. Furthermore, there was a significant difference in the time spent with swimming regarding the effect of age (F(1,41) = 9.147; *p* = 0.00) and the genotype (F(1,41) = 8.668; *p* = 0.01) (Figure 3A). More precisely the AD model-mice swam longer than controls.

#### 2.3.2. Tail Suspension Test (TST)

In the percentage of time spent in the immobile posture, there was no significant difference between the groups with respect to genotype or age (age: F(1,43) = 2.41; *p* = 0.13, genotype: F(1,43) = 1.62; *p* = 0.21, age*genotype: F(1,43) = 0.05; *p* = 0.83; Figure 3B). In the latency of immobility, there was a significant difference between groups (age*genotype: F(1,43) = 5.972; *p* = 0.02; Figure 3C). A post-hoc analysis showed that the difference only occurred in the 8-month-old mice (4-month-old control vs. 3xTg-AD: *p* = 0.16; 8-month-old control vs. 3xTg-AD: *p* = 0.05). The 8-month-old control mice gave up active movement earlier than the 4-month-old ones, while in the 3xTg-AD mice immobility latency was increased at 8-month-old.

#### 2.3.3. Sucrose Preference Test (SPT)

Both genotypes preferred the sucrose solution over water in both age groups (Figure 3D). It was reflected by a higher than 50% sucrose preference value in each group (*p* = 0.00 in each single-sample *t*-test against 50). However, there was no difference between genotypes or ages.

#### 2.3.4. Splash Test

There was a significant difference between the genotypes in the time spent with grooming in both age groups (F(1,40) = 10.972; *p* = 0.00), with significantly less time spent grooming in the 3xTg-AD groups (Figure 3E). Furthermore, the 3xTg-AD mice spent more time in an immobile posture than the controls (F(1,41) = 64.118; *p* = 0.00; data not shown), which effect was more pronounced in the 4-month-old than in 8-month-old animals (age: F(1,41) = 12.629; *p* = 0.00; age*genotype: F(1,41) = 12.325; *p* = 0.00).

#### 2.3.5. Learned Helplessness (LH) Test

Escape latency showed slightly significant differences between genotypes (genotype: F(1,40) = 3.93; *p* = 0.05): the 3xTg-AD animals escaped quicker (Figure 3F). There was a surprising decrease in escape latency with advancing age: it was significantly lower in the 8-month-old animals compared with 4-month-old ones (age: F(1,40) = 6.8; *p* = 0.01).

### 2.4. Acetylcholinesterase (AChE) Immunohistochemistry

Integrated fiber density was lower in 3xTg-AD compared to the control mice (age: F(1,17) = 4.65; *p* = 0.05, genotype: F(1,17) = 39.06; *p* = 0.00; age*genotype: F(1,17) = 5.94; *p* = 0.03; Figure 4A,B). A post-hoc analysis showed that there was a significant difference in 3xTg-AD vs. the control in both age groups with more pronounced effects in the 8- (*p* = 0.00) than in 4-month-old (*p* = 0.02) groups.

### 2.5. Dexamethasone Suppression Test

Elevated corticosterone levels were measured in the 3xTg-AD animals after dexamethasone treatment compared to controls (age: F(1,34) = 0.01; *p* = 0.93; Figure 4C). In the 4-month-old mice, the 3xTg-AD animals had higher levels suggesting dexamethasone non-suppression (genotype: F(1,34) = 2.87; *p* = 0.10; age*genotype: F(1,34) = 3.19; *p* = 0.08). There was no significant difference in adrenocorticotropin (ACTH) levels in either groups (age: F(1,34) = 0.00; *p* = 0.99, genotype: F(1,34) = 0.03; *p* = 0.87, age*genotype: F(1,34) = 0.01; *p* = 0.93; Figure 4D).

## 3. Discussion

Table 1 summarize the results of our experiments and their possible interpretation.

### 3.1. Cognitive Tests

First, we aimed to confirm the appearance of cognitive decline in our local mouse strain. Using the well-known gold standard MWM, we confirmed progressive cognitive decline, which was not apparent in 4-month-old animals, i.e., before the appearance of the pathological phenotype described in the literature [5,6]. The youngest age group with AD-like pathology is usually 6-month [15]. According to the distance traveled in the MWM, the locomotor activity of the 3xTg-AD mice and controls did not differ at any age, i.e., none of the genotypes was inhibited by physical disability.

Next, we introduced SD, as social memory problems might be an important, albeit often neglected aspect of AD. In our experimental design, short-term SD was evaluated, and we found poor short-term social memory both in 4- and 8-month-old 3xTg-AD mice compared with controls. As olfactory dysfunction may occur in this strain only at 1 year, this could not have influenced our outcome [16]. Previous studies using the three-chamber social recognition test, found no significant difference between genotypes in 6-month-old males [17], despite the fact that in the cited experiment mice were given twice as much time to investigate the stimulus animals. Furthermore, we randomized the localization of the familiar mouse, thus, in the cited article social recognition might have been confounded by place preference.

Interestingly, human AD patients also suffer memory loss from recent events first: in a longitudinal study visual short-term memory impairment has been found in familial AD patients years before diagnosis [18]. Thus, our results confirm the behavioral similarity of the 3xTg-AD animal model to that of human AD patients.

### 3.2. Anxiety-like Behavior

In contrast to the MWM, in the OF test, a significant genotype difference in activity was observed; the 3xTg-AD mice moved less, being more immobile than their control counterparts at both ages. In our opinion, this difference might not be functional, but rather a motivational problem. The only escape from the aversive situation in the MWM is to seek dryness, therefore, the drive for movement in this test is strong. In OF, however, the animal’s curiosity about getting to know the environment competes with staying next to a safe wall or freezing into anxiety-induced immobility. Therefore, the lack of active movement and increased immobility seen in the OF test (as well as in NSF and splash tests) may be an anxiety-like symptom as suggested earlier by [19]. Alternatively, the 3xTg-AD mice might have a lower exploratory drive, or a combination of these two factors is also conceivable. Of note, the classical anxiety parameter of OF, the time spent in the center, did not differ between genotypes. The results of others, such as Zhang et al. [20], confirmed significant anxiety in 3xTg-AD animals using even the classical parameters without a locomotor difference. It was possible that the 2.7 times higher box they used gave a greater feeling of security than in our case, thus, our animals were freezing instead of moving to the wall. Additionally, our 3xTg-AD mice spent much more time with self-grooming and immobility depicting their anxiety-like phenotype at both studied ages, which were correlated with cognitive symptoms.

In the EPM test, no significant difference was found between genotypes. In a longitudinal study, Pairojana et al. also described no difference between male 3xTg-AD mice and controls in time spent in the open arm in 3-, 6-, 9-, and 12-month-old animals [21]. Furthermore, the female 3xTg-AD mice spent even more time in the open arm in their experiment. Although another study found anxiety-like behavior in 9-month-old 3xTg-AD mice, the results were overall inconclusive, and this test was not recommended for the 3xTg-AD mice [22].

The results of the LD test seem to be contradictory, too, as 3xTg-AD mice spent more time in the light, the anxiogenic environment. However, similarly to the OF, this can be explained by their higher immobility, rather than less anxiety phenotype. This was further supported by their lower frequency and higher latency to enter the dark compartment. Although a 2013 study found no difference in time spent in a light compartment between genotypes studying 20-month-old mice, they also confirmed the lower frequency of entering the dark compartment [23]. In another study, less time spent in a bright compartment and lower latency to enter the dark compartment was already found in 6-month-old 3xTg-AD mice [20]. Besides age differences, there were substantial differences in the experimental layouts, which may lead to conflicting results (e.g., light and dark boxes used in our experiment were way higher than in [23] or [20]; moreover, contrary to our experiment, mice were put first into the dark compartment, which might determine the place of immobility).

The NSF results could also be interpreted along the same line, as here increased immobility was also detected. Although the enhanced eating time in 3xTg-AD mice seems to be contradictory, our subsequent studies found that 3xTg-AD mice work for food with higher motivation than the controls, which might be explained by metabolic differences (for further interpretation see our subsequent studies).

In addition, progressive decrease in social behavior was observed in 3xTg-AD in SI test, which could also be interpreted as either anxiety-like behavior, or decreased exploratory drive. Decreased social interaction has already been described in 9- [24], 12- [25], and 14-month-old 3xTg-AD animals compared to control, supporting age-related decline. Although a 2012 study led to conflicting results: 12-month-old 3xTg-AD males showed no difference in social interaction compared to controls [26], and at 18 months of age, they showed increased social interest [27]. Different experimental conditions or different animal strains used as controls may cause such a drastic difference, which should be taken into consideration when planning the experiment.

### 3.3. Depression-like Behavior

A significant difference was not found between genotypes in the TST in the main parameter of depression-like behavior (percentage of immobility). However, in the FST, both 4- and 8-month-old 3xTg-AD mice swam more and floated less than controls. A 2014 study found the same results in 12-month-old 3xTg-AD mice, which was—based upon other behavioral tests—interpreted as a sign of general hyperlocomotion [28]. However, our local colony showed hypo- rather than hyperlocomotion in line with other authors (e.g., [21,27,29,30]). In addition, the temperature of the water may also have strongly influenced the outcome [31]. As a previous study indicated some thermoregulatory deficits in 15-month-old 3xTg-AD mice [32]; we might assume that in our case the 3xTg-AD mice were not able to cope with the relatively cold water (24 +/− 1 °C), thus, were more active to maintain core temperature.

In the SPT, Wang et al. found significantly lower sucrose preference in 9-month-old 3xTg-AD than in age-matched control mice (sucrose preference: ~60% vs. ~62%, respectively) [22]. A similar result was found in our experiment in 8-month-old mice (3xTg-AD sucrose preference: ~79%, control: ~86%); however, in our case, the difference did not reach significance. Nevertheless, this tendency might support the age-dependent progression of depressive-like co-morbidity in the 3xTg-AD animals [33].

In the splash test, lack of self-care was monitored as a manifestation of depressive-like behavior. We found that the 3xTg-AD mice were not motivated to keep their fur clean, and in both age groups, the 3xTg-AD animals spent less time grooming compared with their control counterparts. The splash test is an unfairly unpopular test: if we searched for it with the keywords “splash test” and “depression” on PubMed, there were only 132 results. It has not yet been used in an experiment with 3xTg-AD mice. Other tests measuring depression-like parameters in connection with self-care have shown differences between AD and control genotypes at the age of 6-month [17]. In the so-called coat state scores, male 3xTg-AD mice showed a more deteriorated coat state compared to controls. This result partially supported our findings.

Interestingly, the 3xTg-AD and older mice escaped quicker than the controls or younger ones in the LH test. It was definitely not due to memory problems as memory decline should lead to enhanced rather than reduced escape latency. It was possible that the fight-or-flight response has some genotype- and age-dependent factors. Thus, AD or older individuals may be more likely to choose “flight”, but no such data were available in the literature, nor on the LH test in this genetic AD model. Depressive-like behavior was not confirmed in this test.

### 3.4. Acetylcholinesterase (AChE) Immunohistochemistry

A progressive decrease of AChE-positive fiber density in the somatosensory cortex of the 3xTg-AD mice was confirmed, being significantly different between genotypes in both age groups. Indeed, basal forebrain cholinergic abnormalities have also been found in the literature: early cytoarchitectural abnormalities in the basal forebrain were detected by a diffusion MRI in 3xTg-AD mice after 8 months of age [34]. Our results first described the appearance of this AD phenotype at the molecular level at 4 months of age.

### 3.5. Dexamethasone Suppression

Dexamethasone non-suppression was found in 4- but not 8-month-old 3xTg-AD mice, preceding the signs of spatial memory decline. Similar to our results, 4-month-old 3xTg-AD mice reacted to stress with an exaggerated corticosterone rise; however, it was detected in females only [17,35]. The 15-month-old male, but not female, 3xTg-AD mice had higher resting corticosterone levels than their controls [36]. Although a more detailed analysis of the HPA axis of 3xTg-AD mice is needed [37], our data supported comorbidity and the fact that the appearance of depression-like changes could precede the cognitive decline [17].

## 4. Materials and Methods

### 4.1. Animals

The experiments were conducted with 4- and 8-month-old male 3xTg-AD and age-matched C57BL/6J animals. Breeder mice were obtained from the Mutant Mouse Resource and Research Center at the Jackson Laboratory (Bar Harbor, ME, USA, No. 004807) [5]. The local colony was kept at the Institute of Experimental Medicine, Budapest, Hungary. Mice were housed in standard mouse cages (37.3 × 23.4 × 14.0 cm; Charles River, Veszprém, Hungary) maintained in a standard environment (20+/−2 °C temperature; 60+/−10% humidity; 12-h light/dark cycles, lights on at 7 p.m.). The experiments were conducted during the dark cycle which is the active period of the mice. Food (standard mice-pellet, Charles River, Veszprém, Hungary) and tap water were available ad libitum. Three–five littermates were kept together, and their environment was enriched with a paper roll and nest building materials. All tests were approved by the local committee of animal health and care and performed according to the European Communities Council Directive recommendations for the care and use of laboratory animals.

### 4.2. Behavioral Tests

Behavioral tests were video recorded and analyzed later either by Noldus Ethovision (13.0, Wageningen, The Netherlands) (MWM, OF, EPM, and NSF locomotion) or by Solomon Coder (SD, OF: anxiety-like behavior, LD, NSF: eating, and anxiety-like behavior, SI, FST, TST, splash) by an experimenter blind to the treatment groups. The tests were done during the active, subjective dark phase of the animals, most of them under red light, however, some (MWM, EPM, SI, FST, TST, and splash) were performed under normal lighting. For the OF, LD, and NSF video recordings of four animals were simultaneously made, and boxes were placed in the 2 × 2 position. The test boxes (where applicable) were cleaned between animals with ethanol. A behavioral test battery was used with 4 different cohorts (Table 2, Figure 5).

#### 4.2.1. Cognitive Tests

##### Morris Water Maze (MWM) Test

This test measures spatial memory ([38], Figure 6A). The same mice were tested repeatedly for five (4-month) or four (8-month, without habituation) consecutive days, five times per day. Mice had to swim in a circular-shaped pool and find an underwater platform using objects placed in the room as spatial cues, as described earlier by [38]. The last, either the sixth or fifth day of the test was the “probe” test, during which the platform was removed, and the animals were allowed to swim for 60 s. In this case, the total distance traveled-reflecting locomotion-was measured.

##### Social Discrimination (SD) Test 

SD consisted of four phases: open field (OF) (e.g., habituation to the arena, see Section 4.2.2 OF), habituation (to two small containers with grid), sociability, and SD, according to previous description [39] (Figure 6B). During the SD phase frequency and time spent with each conspecific were measured. Any other type of behavior was labelled as “other”. The DI index was calculated based on this equation:DI = (t_‘old’_ − t_‘new’_) − (t_‘old’_ + t_‘new’_) × 100
where t_‘old’_ stood for the time spent with sniffing the cage containing the familiar stimulus mouse during SD phase; t_‘new’_ stood for the time spent with sniffing the cage containing the unfamiliar stimulus mouse in SD phase.

#### 4.2.2. Tests of Anxiety-Like Behavior

##### Open Field (OF) Test

OF test was applied to measure locomotor activity as well as the anxiety-like behavior (Figure 7A). Mice were placed into an empty white plastic box (40 cm × 36 cm × 15 cm) without bedding for 5 min. Behavioral elements assessing locomotion and anxiety parameters (time spent in centrum) were analyzed. Duration and frequency of exploration, immobility, rearing, grooming, and total distance travelled were also measured.

##### Elevated Plus Maze (EPM)

The EPM was performed according to the protocol previously described [40]. The locomotor independent anxiety is expressed as a percentage of open arm entries versus all arm entries (Figure 7B).

##### Light-Dark (LD) Test

The LD box had two compartments: a light (white colored) that was open from above and a dark (black colored) that was closed from every side (40 × 20 × 25 cm each compartment) except for the small gate (5 × 5 cm) where animals could freely pass between the two (Figure 7C). This test typically measures anxiety-like behavior similarly to OF and EPM tests (based upon fear of open spaces [41]). Less anxious mice spend more time in the light compartment, so during this 10-min-long test, the duration of time spent in each compartment was measured.

##### Novelty Suppressed Feeding (NSF) Test

During NSF, the latency to approach the food and eating it shows how the animal copes with a behavioral conflict that is inversely related to anxiety-like symptoms caused by a novel environment [42,43]. Mice were weighed, and all food was removed from their cages 24 h prior to the test; water was freely available. During the test, mice were placed in an empty white plastic box (40 cm × 36 cm × 15 cm) without bedding (Figure 7D). A small piece of a mouse pellet was placed in the center of the arena, and each mouse was placed in the corner. During the 6 min observation period latency as well as the duration of eating was measured. The weight of the pellet and mice was measured before and after the test.

##### Social Interaction (SI) Test

SI test consisted of a 2 × 10 min habituation phase and a 10-min-long interaction on the following day (Figure 7E) as previously described [40]. However, in our case, the habituation was repeated after 6 h, with the order of pairs reversed and keeping the same bedding.

#### 4.2.3. Tests of Depression-like Behavior

##### Forced Swim Test (FST)

FST is one of the most popular methods to measure depressive-like behavior in rodents (Figure 8A). The test was performed according to the protocol described by [39]. The immobility or floating behavior reflects the behavioral despair, often found in depressed human patients.

##### Tail Suspension Test (TST)

TST is considered the “dry” version of FST ([44], Figure 8B). Mice were hung for 6 min with a 10 cm long leucoplast tape which was fixed 1 cm from the end of their tails. A 3 cm long plastic cylinder was pulled over their tail, so they were not able to turn and climb back on their own tail. From this position, the animals could escape, however, they might show escape-oriented behaviors, and in the absence of this behavior, the immobility is taken as behavioral despair.

##### Splash Test

Mice were placed into an empty white plastic box (11 cm x 38 cm x 15 cm) without bedding (Figure 8C). First, the animals were habituated to the arena for 10 min. After mice were splashed on their back with 10% sucrose solution with Pasteur pipettes, the duration of grooming (self-caring) and immobility behavior was measured for 5 min. Animals developing depressive-like symptoms spent less time with grooming and were more immobile [45,46].

##### Sucrose Preference Test (SPT)

SPT is meant to measure anhedonia (inability to feel joy) (Figure 8D). The animals’ preference for sugar water over normal tap water was tested by measuring the amount of liquid consumed daily. Prior to the test, the mice were separated in normal mouse cages and were accustomed to the taste of sugar water for 3 days. Then mice got two similar water bottles: one with sucrose solution (1%) and one with normal tap water. The weight of these bottles was measured every day for 3 consecutive days. Mice developing anhedonia do not discriminate between the two types of water, while a healthy mouse usually prefers sucrose solution over tap water [43,47]. Sucrose preference was calculated as follows: (sucrose consumption)/(sucrose + water consumption) × 100 Learned Helplessness (LH) Test.

The LH test was conducted in an automated shuttle box apparatus, which had two identical compartments with photobeam sensors, stimulus light, tone generator, stainless steel grid floor, and a guillotine door (Figure 8E) as described earlier [38]. The active avoidance (escape tries) was measured [48]. Animals that had developed LH did not avoid the shocks even when they could do so. The test was analyzed by the built-in program of the shuttle boxes’ computer.

### 4.3. Immunohistochemistry

Animals from cohort 2 were anesthetized intraperitoneally with a ketamine-xylazine solution and transcardially perfused with phosphate-buffered saline (PBS) followed by 4% paraformaldehyde. Their brains were kept in a PBS-azide solution until processing. Then 30 μm coronal slices were prepared.

#### 4.3.1. Cholinergic Fiber Labelling in the Somatosensory and Motor Cortex

AChE histochemistry with silver nitrate intensification was performed to label and visualize cholinergic fibers in the somatosensory and motor cortex as described earlier [49,50,51]. Brain sections were incubated in a buffer containing: sodium acetate buffer (0.1 M; pH 6), acetylthiocholine-iodide (0.05%), sodium citrate (0.1 M), copper sulphate (0.03 M), and potassium ferricyanide (5 mM). This was followed by ammonium sulfide (1%) and then silver nitrate (1%) incubation for 1 min to visualize AChE-positive neurons and fibers.

#### 4.3.2. Analysis of Histological Data

From each animal 6 sections from Bregma 0.49 mm to Bregma −1.23 mm with a 120 μm inter-sectional distance were selected and analyzed for AChE fiber density under an Olympus BX51 microscope (Budapest, Hungary). Using ImageJ software (Bethesda, MD, USA) after background subtraction and grey scale threshold determination, the surface area density of the cortical AChE-positive fibers was measured. The integrated fiber density was measured in the same area for each section between the 4th and 5th cortical layers (Figure 9).

### 4.4. Dexamethasone Suppression Test

In Cohort 4 mice were treated with dexamethasone (0.1 mg/kg ip, Oradexon, Organon, Oss, Hollandia) during the early morning (at the time of the beginning of the dark phase or active period of the animals, with the highest ACTH-corticosterone level) [53]. Trunk blood was collected 6 h after ip injections in ice-cooled Eppendorf tubes. Serum was separated by centrifugation (3000 rpm for 10 min at 4 °C) and stored at −20 °C until assayed. The ACTH and corticosterone concentrations were determined by radioimmunoassay in duplicates as described earlier [54].

### 4.5. Statistical Analysis

Data were analyzed by factorial or mixed (two-way × repeated measures) ANOVA with StatSoft Statistica 13.0 (Tulsa, OK, USA). Post-hoc comparison was made by the Fisher LSD test. For SD as well as for SPT a single sample *t*-test against 0 or 50% as chance levels was also conducted. Data are presented as mean ± S.E.M. The level of statistical significance was set at *p* < 0.05 in all statistical analyses.

## 5. Conclusions

Our study examined the comorbidity of cognitive decline in AD with anxiety- and depression-like behavior, examining behavioral alterations using an extended test battery in 3xTg-AD mice models before (4-month) and after (8-month) the onset of AD pathology with unique details. Short-term memory loss (measured in SD) occurred early, consistent with human data. However, long-term memory impairment (measured in MWM), and fiber-density loss (measured with AChE immunohistochemistry) in the cortex showed age-dependent progression. At the same time, several anxiety- and depression-like alterations were already characteristic of 3xTg-AD animals at 4 months of age, including enhanced immobility in OF, LD, and NSF tests, novel environment-induced self-grooming in OF and splash tests, and dexamethasone non-suppression. These suggested the appearance of an anxiety-like phenotype even before the onset of AD pathology. The onset of social anxiety (measured in SI)—similar to spatial memory decline—showed age-dependent progression. Of note, some of these results could also be explained by a possible decrease in exploratory drive in the 3xTg-AD mice, also suggested by a decreased locomotion in other tests. In summary, we showed that some behavioral characteristics of 3xTg-AD mice might be considered as anxiety- and depressive-like; therefore, these selected tests might be used to study comorbidity. We further refined our previous knowledge of the 3xTg-AD mouse model and provided a well-characterized experimental tool for testing new treatment options.

## Figures and Tables

**Figure 1 ijms-23-10816-f001:**
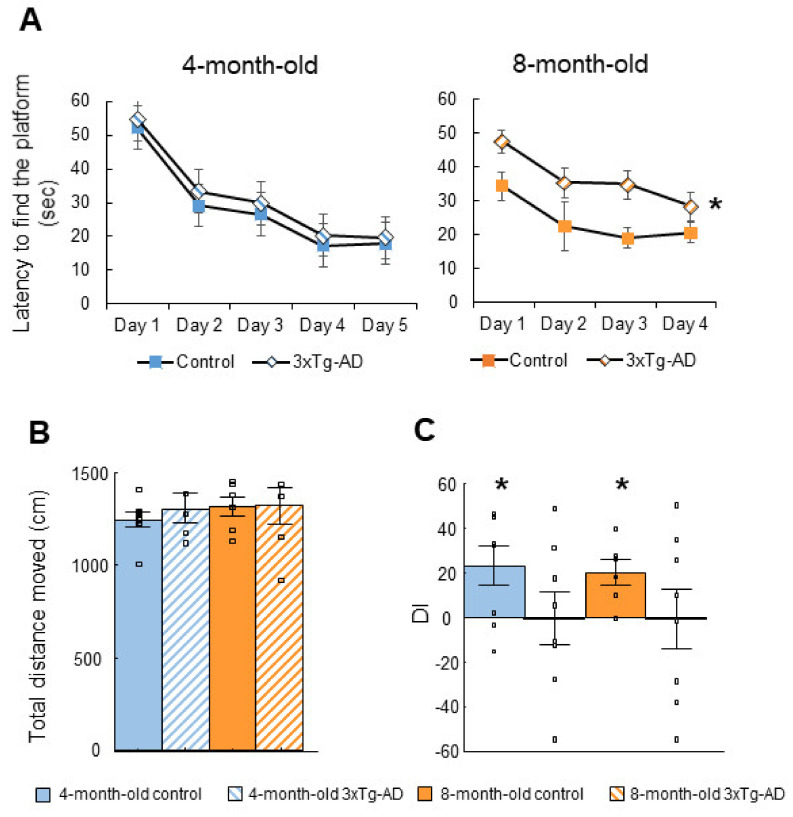
Cognitive behavior. A-B: The Morris Water Maze (MWM) test. (**A**): 3xTg-AD mice found the platform slower at four months of age than their age-matched control counterparts with a genotype difference tendency only at day 5 (*p* = 0.06). The genotypic difference increased further at 8 months of age (* *p* = 0.00). (**B**): Results of the MWM probe test. There was no significant difference in the total distance moved (*p* < 0.05). (**C**): The social discrimination (SD) test. The discrimination index (DI) of the 3xTg-AD mice was not significantly different from zero (4-month-old: *p* = 0.99; 8-month-old: *p* = 0.98). In contrast, control group had intact social memory throughout (4-month-old: * *p* = 0.03; 8-month-old: * *p* = 0.02).

**Figure 2 ijms-23-10816-f002:**
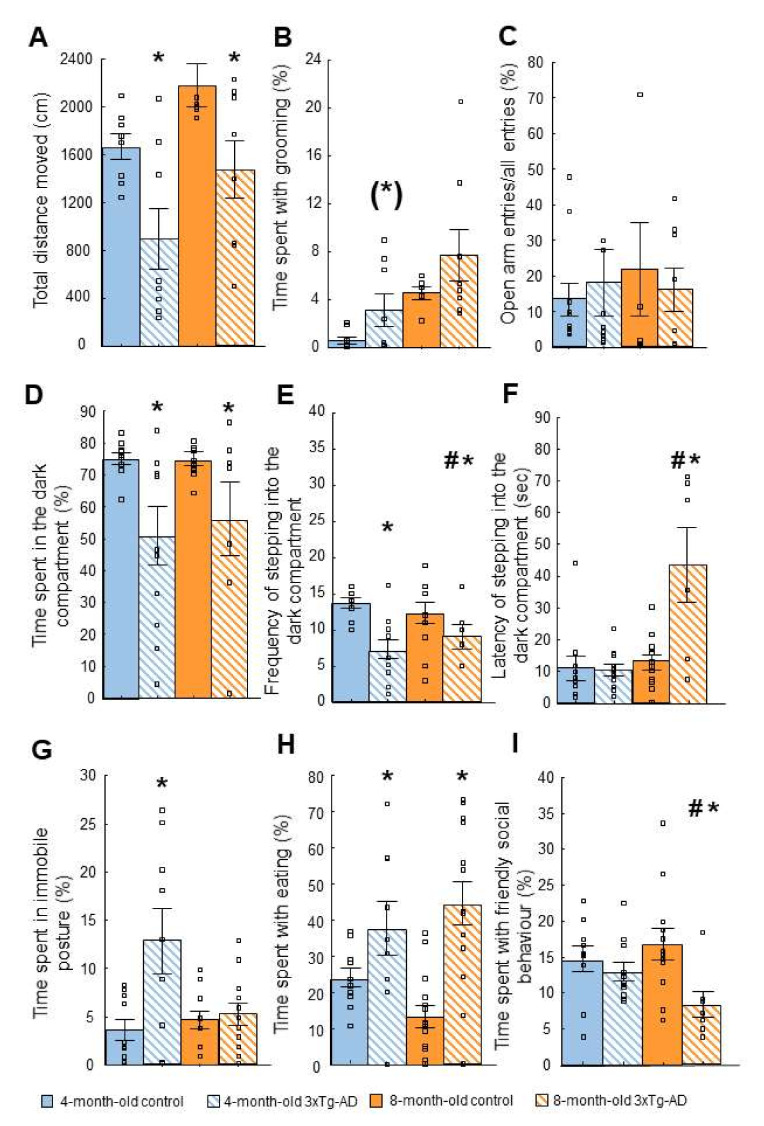
**Anxiety-like behavior.** (**A**,**B**): Open field (OF) test. (**A**): There was a significant difference in time spent in immobile posture: 3xTg-AD mice spent significantly less time with active movement and more with immobility compared to control both in 4- (* *p* = 0.02) and in 8-month (* *p* = 0.04) groups. (**B**): 3xTg-AD animals in both age groups spent more time with self-grooming than the controls, however, the difference was only marginally significant ((*) *p* = 0.09). For 8-month group, this difference was further reduced (*p* = 0.24). (**C**): Elevated plus maze (EPM) test. There was no significant difference between genotypes either in the classical parameters (open arm time, data not shown), or in locomotion-independent anxiety parameter, the % of open arm frequency. (**D**–**F**): Light-dark (LD) test. (**D**): Significant difference was found between genotypes in percentage of time spent in the dark compartment of the box (genotype: * *p* = 0.00). (**E**): The frequency of stepping into the dark compartment was also significantly lower in the 3xTg-AD mice compared to the controls (genotype: * *p* = 0.00). (**F**): There was also a significant difference in the latency of stepping into the dark compartment in case of the 8-month-old animals, 3xTg-AD mice entered later than the controls (age*genotype: * *p* = 0.04; 8-month-old post-hoc: # *p* = 0.01). (**G**–**H**): Novelty suppressed feeding (NSF) test. (**G**): There was a significant difference in the time spent with immobility between genotypes in both age groups (genotype: * *p* = 0.03). More precisely, 4-month-old 3xTg-AD mice spent more time with this behavior than their controls, which suggested enhanced anxiety-like behavior. (**H**): 3xTg-AD mice spent more time with eating without age difference (genotype: * *p* = 0.00). (**I**): Social interaction (SI) test. There was a significant difference between genotypes in the time spent with social behavior in case of the 8-month-old animals with age difference (age*genotype: * *p* = 0.05; 8-month-old post-hoc: # *p* = 0.01). The 3xTg-AD mice spent less time with friendly social behavior compared to the controls.

**Figure 3 ijms-23-10816-f003:**
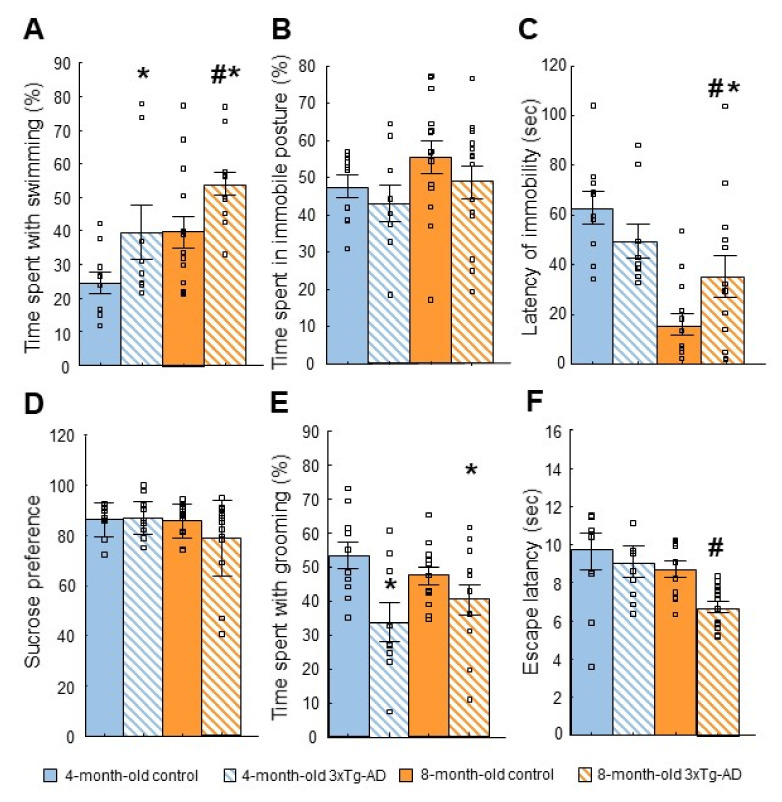
Depressive-like behavior. (**A**): Forced swim (FST) test. There was a significant difference in the time spent swimming regarding the effect of genotype (* *p* = 0.01) and age (# *p* = 0.00). (**B**,**C**): Tail suspension (TST) test. (**B**): There was no significant difference between the groups in the time spent in immobile posture (*p* > 0.05). (**C**): In the case of the latency of immobility, there was a significant difference between genotypes (age*genotype: *# *p* = 0.02). (**D**): Sucrose preference test (SPT). Both genotypes preferred sucrose solution over water in both age groups, there was no significant difference between groups. The sucrose preference value differed from random 50% in each case (*p* = 0.00 in each single-sample *t*-test against 50). (**E**): Splash test. There was a significant difference between the genotypes in the time spent grooming in both age groups (* *p* = 0.00) with significantly less time spent grooming in the 3xTg-AD groups. (**F**): Learned helplessness (LH). Escape latency was lower in 3xTg-AD than in the control mice (*p* = 0.05). There was a surprising decrease in escape latency with advancing age: it was significantly lower in 8-month-old animals compared with 4-month-old ones (age: # *p* = 0.01).

**Figure 4 ijms-23-10816-f004:**
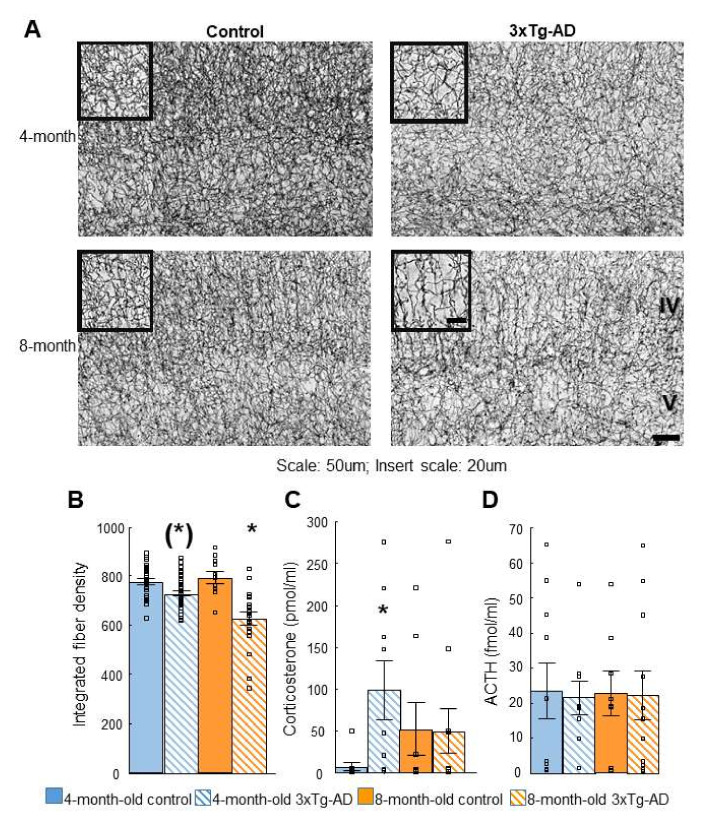
Visual (**A**) and quantitative (**B**) results of the acetylcholinesterase (AChE) immunohistochemistry. The integrated fiber density was measured in the same area for each section between the 4th and 5th cortical layers (see Figure 4). Post-hoc analysis showed that in 8-month-old animals the difference was more pronounced (4-month: * *p* = 0.02; 8-month: * *p* = 0.00). (**C**): Elevated corticosterone levels were measured in 3xTg-AD animals after dexamethasone treatment compared to controls, however, this difference was only significant in 4-month-old animals (* *p* < 0.05). (**D**): There was no significant difference in adrenocorticotropin (ACTH) levels in either group (*p* > 0.05).

**Figure 5 ijms-23-10816-f005:**
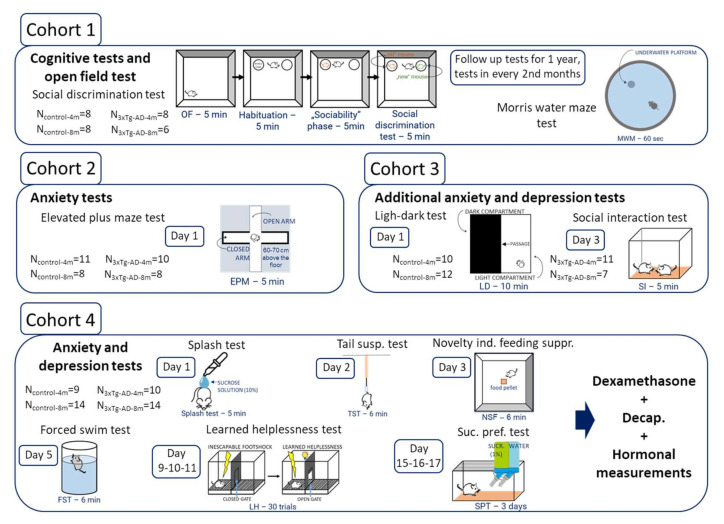
The experimental design in different cohorts.

**Figure 6 ijms-23-10816-f006:**
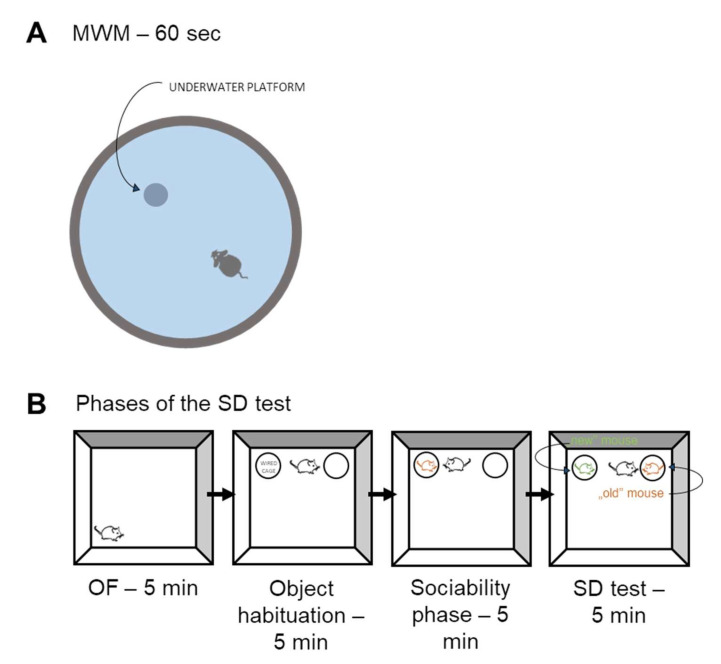
Schematic representation of behavioral tests measuring cognitive functions. (**A**): Morris water maze (MWM) test. Mice were tested for five (4-month-old) or four (8-month-old) consecutive days five times per day. Mice had to swim in a circular shaped pool and find an underwater platform using external spatial cues. (**B**): Social discrimination (SD) test consisted of four phases: open field (OF) test (e.g., habituation to the arena); habituation (to two small containers with grid), sociability, and social discrimination. For the sociability phase, under one of the containers one juvenile conspecific was placed (‘old’). For the next phase, an additional conspecific (‘new’) was placed under the other container. The position of the ‘old’ mouse was interchanged compared to the previous phase to avoid place preference. The test animal could freely behave for 5 min during each phase.

**Figure 7 ijms-23-10816-f007:**
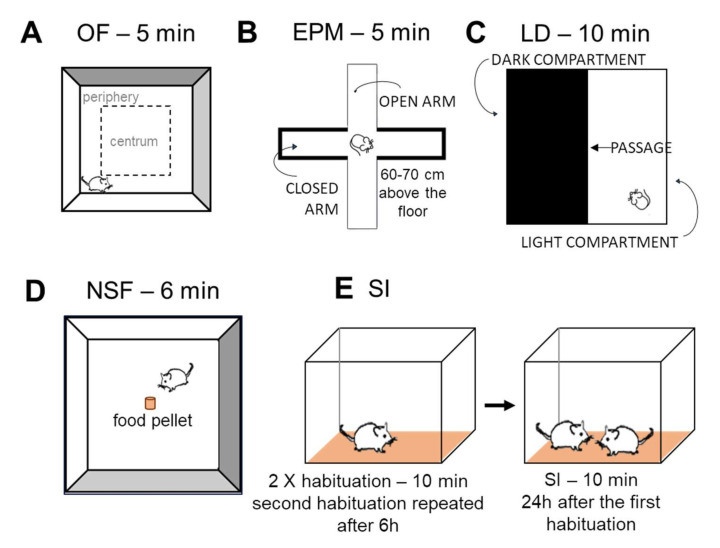
Schematic representation of behavioral tests measuring anxiety-like behavior. (**A**): Open field (OF) test (the first phase of SD). Animals had 5 min to explore the new environment. (**B**): Elevated plus maze test (EPM). Mice were put into the center of EPM, which had open and closed, highly walled parts. The test lasted 5 min. (**C**): Light-dark (LD) box. Mice were put into the light part of the box. There was also a dark compartment with black walls and cover. The two compartments were passable through a small gate. LD test lasted 10 min. (**D**): Novelty suppressed feeding (NSF). Sniffing and eating food in a novel environment after a 24 h period of food deprivation was measured for 6 min. (**E**): Social interaction (SI) test. After habituation to the test cage, mice were placed into a plexiglass aquarium with bedding at the same time and were then tested for 10 min for the following parameters: friendly, aggressive, or defensive social behavior, and other non-social behavior.

**Figure 8 ijms-23-10816-f008:**
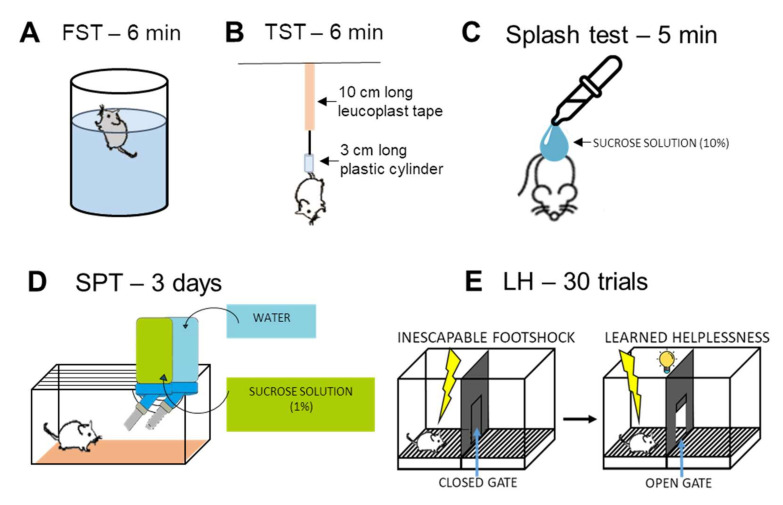
Schematic representation of behavioral tests measuring depressive-like behavior. (**A**): Forced swim test (FST). Mice are forced to swim in a narrow, transparent cylinder filled with tap water from which they cannot escape for 6 min. (**B**): Tail suspension test (TST). Mice were hung for 6 min with a 10 cm long leucoplast tape which was fixed 1 cm from the end of their tails. A 3 cm long plastic cylinder was pulled over their tail, so they were not able to turn and climb back on their tail. (**C**): Splash test. After 10 min habituation mice were splashed with 10% sucrose solution with Pasteur pipettes and then for 5 min the duration of grooming (self-caring) behavior was measured. (**D**): Sucrose preference test (SPT). Mice should discriminate between tap water and 1% sucrose solution. Healthy mice usually prefer sucrose solution over tap water. (**E**): Learned helplessness (LH) test. In both phases, an initial 5 min habituation period preceded the first trial. Test phase consisted of 30 trials of escapable foot-shock (0.15 mA intensity, 10 s maximum duration, 30 s average intertrial interval) with the door open from the light onset.

**Figure 9 ijms-23-10816-f009:**
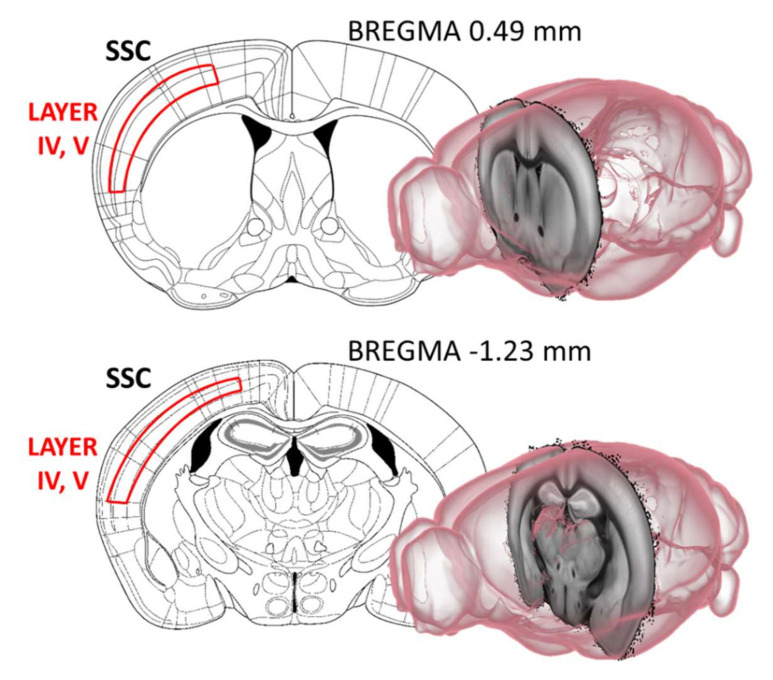
Acetylcholinesterase (AChE) fiber density was investigated between the 4th and 5th cortical layers of the somatosensory cortex (SSC). From each animal 6 sections from Bregma 0.49 mm to Bregma −1.23 mm, with 120 μm inter-sectional distance were selected and analyzed for AChE fiber density. Images were created with the help of [52].

**Table 1 ijms-23-10816-t001:** Summary and interpretation of results. Arrows are used to indicate significant increase (↑) or decrease (↓) in a given parameter in 3xTg-AD animals compared to controls. Abbreviation: (4-m): only four-month-old animals; (8-m): only eight-month-old animals.

Test	Main Finding in 3xTg-AD Mice Compared to Controls	Interpretation
Morris water maze (MWM)	↑ Latency to find the platform (8-m)	Progressive long-term memory loss
Social discrimination (SD)	3xTg-AD DI did not differ from 0	Short-term memory impairment
Open field (OF)	↓ Time with active movement	Anxiety-like behavior
Elevated plus maze (EPM)	-	-
Light-dark box (LD)	↓ Motivation to explore	Anxiety-like behavior
Novelty suppressed feeding (NSF)	↑ Immobility	Anxiety-like behavior
Social interaction (SI)	↓ Social behavior (8-m)	Progressive anxiety-like behavior
Forced swim test (FST)	↑ Swimming	Stress coping impairment?
Tail suspension test (TST)	-	-
Sucrose preference test (SPT)	-	-
Splash	↓ Grooming	Depression-like behavior
Learned helplessness (LH)	↓ Escape latency	Stress-coping impairment?
Dexamethasone suppression	Non-suppression (4-m)	Depression-like behavior
AChE immunohistochemistry in cortex	↓ Fiber density (8-m)	Progressive fiber density decrease

**Table 2 ijms-23-10816-t002:** Tests and sample sizes.

Test	4-Month		8-Month	
	3xTg-AD	Control	3xTg-AD	Control
Morris Water Maze (MWM)	8	8	6	8
Social discrimination (SD)	8	8	6	8
Open field (OF)	8	8	6	8
Elevated plus maze (EPM)	10	11	8	8
Light-dark box (LD)	11	10	7	12
Novelty suppressed feeding (NSF)	8	10	10	14
Social interaction (SI)	11	10	7	12
Forced swim test (FST)	8	10	13	14
Tail suspension test (TST)	9	10	14	14
Splash	9	10	12	13
Sucrose preference test (SPT)	9	10	14	14
Learned helplessness (LH)	9	10	14	11

## Data Availability

The data presented in this study are available on request from the corresponding author.

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
