# Peer review of "Investigation of Anxiety- and Depressive-like Symptoms in 4- and 8-Month-Old Male Triple Transgenic Mouse Models of Alzheimer’s Disease"

_ijms, 2022, doi:10.3390/ijms231810816_

Round 1

Reviewer 1 Report

The manuscript entitled "Appearance of anxiety- and depressive-like symptoms in 4- and 8-month-old male triple transgenic mice model of Alzheimer's disorder" aims to confirm that the triple-transgenic mouse (3xTg-24 AD) is a good preclinical model of AD and depression co-morbidity. The manuscript is written well in general. The results are valuable and systematic, but their interpretation and discussion are problematic. It seems that the authors had a presumption that depression is a co-morbidity of AD. The results do not refer to that, in my opinion. Many papers recently drew the attention that depression may be the consequence of AD treatment (for example https://www.tandfonline.com/doi/abs/10.1080/17425255.2021.1931681). The authors should take this into account and modify their discussion and conclusions. The results are not strongly indicating their initial hypothesis. Literature should be more up-to-date. There are more recent articles that could give a completely different connotation to the results presented in this manuscript.

Line 46: AD is rather a form of dementia, not a cause. 

Author Response

Thank you very much for your valuable suggestions. We hope that you will find our answer satisfactory.

The manuscript entitled "Appearance of anxiety- and depressive-like symptoms in 4- and 8-month-old male triple transgenic mice model of Alzheimer's disorder" aims to confirm that the triple-transgenic mouse (3xTg-24 AD) is a good preclinical model of AD and depression co-morbidity. The manuscript is written well in general. The results are valuable and systematic, but their interpretation and discussion are problematic. It seems that the authors had a presumption that depression is a co-morbidity of AD. The results do not refer to that, in my opinion.

Answer: We modified our interpretation according to the suggestion and made changes even in the title. The co-morbidity is well confirmed in the literature based upon clinical data (cited also in the reference recommended by the reviewer bellow). However, here we intended to examine whether the 3xTg-AD mouse is a suitable model for studying such co-morbidity, which -we agree- was not convincingly confirmed.

Many papers recently drew the attention that depression may be the consequence of AD treatment (for example https://www.tandfonline.com/doi/abs/10.1080/17425255.2021.1931681). The authors should take this into account and modify their discussion and conclusions.

Answer: Thank you for drawing our attention to this fact. We added the suggested reference and idea into the introduction.

The results are not strongly indicating their initial hypothesis.

Answer: We modified our conclusion and came up with an alternative explanation as well.

Literature should be more up-to-date. There are more recent articles that could give a completely different connotation to the results presented in this manuscript.

Answer: We rewrote the discussion to be more focused and added more up-to-date literature from 2021-22.

 Line 46: AD is rather a form of dementia, not a cause. 

Answer: Corrected.

Reviewer 2 Report

The submitted manuscript has investigated using behavioral, histochemical and hormone-assay approaches several characteristics of the triple-transgenic mouse (3xTg- 24 AD) used as a preclinical model of Alzheimer’s disease.

The paper presents the results of a carefully done study. However, in my opinion, the paper is too verbose. The materials and methods section could be reduced by 50% without decreasing the clarity of the paper. It is not necessary to describe in detail methods used when a reference can be enough.

More details about the quantification of AChE histochemistry and higher quality micrographs are necessary. The material submitted does not support the author’s description.

A discussion focussed on the interpretation of the results obtained without particular speculations will increase the clarity of the paper.

Author Response

Thank you very much for your valuable suggestions. We hope that you will find our answer satisfactory.

The submitted manuscript has investigated using behavioral, histochemical and hormone-assay approaches several characteristics of the triple-transgenic mouse (3xTg- 24 AD) used as a preclinical model of Alzheimer’s disease.

The paper presents the results of a carefully done study. However, in my opinion, the paper is too verbose. The materials and methods section could be reduced by 50% without decreasing the clarity of the paper. It is not necessary to describe in detail methods used when a reference can be enough.

Answer: We made attempt to reduce the description of the method keeping only sentences highlighting the difference from the already described protocol.

More details about the quantification of AChE histochemistry and higher quality micrographs are necessary. The material submitted does not support the author’s description.

Answer: We added some details to the method and corrected the figures. Moreover, we used more references in line with the previous suggestion to reduce the length of the method section.

A discussion focussed on the interpretation of the results obtained without particular speculations will increase the clarity of the paper.

Answer: We reduced the discussion to be more focused.

Round 2

Reviewer 1 Report

The authors improved the manuscript according to the comments. I recommend it for publication in present form. 

Reviewer 2 Report

The authors have answered to my criticisms and the paper was significantly improved.